# Transformation of Resinous Components of the Ashalcha Field Oil during Catalytic Aquathermolysis in the Presence of a Cobalt-Containing Catalyst Precursor

Irek I. Mukhamatdinov [1,*], Indad Sh.S. Salih [1], Ilfat Z. Rakhmatullin [1], Nikita N. Sviridenko [2], Galina S. Pevneva [2], Rakesh K. Sharma [3] and Alexey V. Vakhin [1,*]

[1] Institute of Geology and Petroleum Technologies, Kazan Federal University, 18 Kremlyovskaya Str., 420008 Kazan, Russia; indad.salih@mail.ru (I.S.S.S.); ilfat89rakhmatullin@gmail.com (I.Z.R.)
[2] Institute of Petroleum Chemistry, Siberian Branch of the Russian Academy of Sciences, 4, Akademicheskiy Ave, 634055 Tomsk, Russia; nikita26sviridenko@gmail.com (N.N.S.); pevneva@ipc.tsc.ru (G.S.P.)
[3] Indian Institute of Technology Jodhpur, NH-62, Nagaur Road, Karwad, Jodhpur 342037, India; rks@iitj.ac.in
* Correspondence: iimuhamatdinov@gmail.com (I.I.M.); vahin-a_v@mail.ru (A.V.V.); Tel.: +7-9503-222563 (I.I.M.); +7-9870-010781 (A.V.V.)

**Abstract:** The aim of this work was to study the fractional composition of super-viscous oil resins from the Ashalcha field, as well as the catalytic aquathermolysis product in the presence of a cobalt-containing catalyst precursor and a hydrogen donor. The study was conducted at various durations of thermal steam exposure. In this regard, the work enabled the identification of the distribution of resin fractions. These fractions, obtained by liquid adsorption chromatography, were extracted with individual solvents and their binary mixtures in various ratios. The results of MALDI spectroscopy revealed a decrease in the molecular mass of all resin fractions after catalytic treatment, mainly with a hydrogen donor. However, the elemental analysis data indicated a decrease in the H/C ratio for resin fractions as a result of removing alkyl substituents in resins and asphaltenes. Moreover, the data of $^1$H NMR spectroscopy of resin fractions indicated an increase in the aliphatic hydrogen index during catalytic aquathermolysis at the high molecular parts of the resins R3 and R4. Finally, a structural group analysis was carried out in this study, and hypothetical structures of the initial oil resin molecules and aquathermolysis products were constructed as well.

**Keywords:** heavy oil; resins; fractionation; MALDI; elemental analysis; nuclear magnetic resonance; structural group analysis

## 1. Introduction

Nowadays, the refining industry is tending toward an increase in the mass of the feedstock being processed and this in turn is leading to the development and implementation of additional processes for the preparation of oils based on the destruction of the structures of high molecular mass compounds in the oil, which leads to an increase in the content of light fractions in oil feedstock [1–5]. Furthermore, one of the most common methods for increasing the yield of light fractions is the method of component destruction of heavy oil feedstock (thermal, thermal radiation, etc.). In this regard, it is important to know the structure of resins and asphaltenes, as well as the number and nature of molecular fragments, as these contain heteroatoms that play a significant role in the destruction of high molecular mass compounds [6–10].

However, the problem of loading the feedstock can already be solved at the stage of oil production by using the water-soluble and oil-soluble catalyst precursors used in steam thermal technologies. Catalyst precursors in reservoir conditions have an active form and promote the conversion of oil without significant structural changes in the used equipment [11–17].

Changes in the physicochemical properties of heavy oils and natural bitumen (density, viscosity in reservoir conditions) as a result of various influences are primarily explained by the transformations which their high molecular mass compounds (HMCs)—resins and asphaltenes—undergo. The heteroatomic compounds present in oil are mainly concentrated in them and, in light of this, the transformations of model compounds under thermal steam exposure at various conditions have been widely investigated. Sulfur-containing model compounds of different classes demonstrate different abilities to generate hydrogen sulfide under conditions simulating the process of cyclic thermal steam exposure (CSS) [18], while the formation of $H_2S$ increases with increasing temperature. However, under supercritical conditions (in the presence of water and other solvents at a temperature of 460 °C), radical mechanisms of desulfurization reactions are involved. It has been shown that thiophenols and n-alkyl and aryl sulfides undergo desulfurization more easily than the cyclic sulfur-containing compounds—thiophenes and benzothiophenes [19,20]—while treatment of 15% aqueous formic acid under supercritical conditions promotes the occurrence of these reactions [19].

A significant increase in the dissociation constant of water with increasing temperature means that water becomes both a stronger acid and a stronger base. Thus, in addition to the normal increase in rate expected with increasing temperature, there is also an increase in rate for both the acid and base catalyzed reactions. This important class of reactions, which require both acidic and basic catalysis, is thus accelerated three times with increasing temperature [21].

Usually, substituents are more reactive than heterocyclic rings of model substances, including sulfur- and nitrogen-containing, monocyclic, bicyclic, and polycyclic aromatic compounds, in which a rupture of bridges connecting aromatic rings has been observed [20]. The cleavage of C-O bonds in ethers and esters was expected under relatively mild conditions in aqueous media. The C-S and C-N bonds were also expected to break quite easily. The wide range of C-C, C-O, and C-N bond formation reactions that have been found to typically occur via ionic mechanisms were not expected. Heterocyclic rings react relatively easily at higher temperatures or in the presence of acids or reducing agents; however, it has been unexpectedly discovered that the ring opening occurs by cleavage of C-C bonds in addition to the more common cleavage of C-S and C-N bonds [20].

In addition to model compound studies, discussions of experiments on the transformation of heavy oil under different conditions can be found in the literature. The change in the component composition of heavy oils depends on their initial composition, but there is a general pattern of decreasing resin and asphaltene content while increasing the content of saturated and aromatic hydrocarbons [22–26]. The authors of [26] showed that the effect of the amount of water on the thermolysis processes of heavy oil accordingly depends on the initial composition of the oil; the water can either promote or limit the processes of denitrogenation and desulfurization. Moreover, in their analysis, the authors suggest that water can promote the oxidative transformation of alcohol–benzene resins into asphaltenes or prevent thermal decomposition of asphaltenes. The authors of [23] showed the influence of rock minerals on the aquathermolysis processes of heavy oil in model experiments in an autoclave in the presence of catalysts and mineral materials, with water vapor at 240 °C for 24 h. They established that the combined use of catalysts and minerals leads to a more significant decrease in viscosity due to a decrease in the contents of resins and asphaltenes in the minerals as a result of their destruction, which was confirmed, in particular, by a decrease in the average molecular mass of the asphaltenes and gas formation: light hydrocarbons, hydrogen, $CO_2$, and $H_2S$ are formed.

The catalytic effect of minerals is attributed by the authors of [22] to the formation, decomposition, and transfer of carbocations on their surfaces, which leads to a change in the distribution of the surface charge of asphaltene molecules. This process is called the "dynamic inductive effect" and contributes to a change in the electron cloud of cationic and ionic bonds. The latter, in turn, leads to easier decomposition of the C–C, C–O, C–S, and C–N bonds and accelerates the aquathermolysis processes of heavy oil, leading to a decrease in

its viscosity and average molecular mass. The steam exposure results in the desulfurization of heavy oil through the release of sulfur, mainly in condensed aromatic nuclei, producing sulfides of transition metals, which also have a catalytic effect and can accelerate the aquathermolysis [22]. Furthermore, the cleavage of the C–S bond in asphaltenes can also lead to the formation of $H_2S$. It has been shown that temperature is one of the main factors that considerably affect the formation of $H_2S$ [25]. Regardless of the composition of oil sands and their sulfur content, the formation of $H_2S$ increased with temperature in aquathermolysis experiments ranging from 240 °C to 320 °C (duration = 203 h, P = 100 bar, $V_{oil} \sim V_{water}$). The authors reached the conclusion that the source of the hydrogen sulfide was resinous–asphaltene components, while sulfur was redistributed into aromatic compounds and insoluble organic residue and released as $H_2S$. The insoluble residue acted as a trap for sulfur initially present in the oil. Moreover, the state of water also had an effect, which means that the formation of $H_2S$ is twice as high when water is in a state of liquid/vapor equilibrium than when it is in a liquid state.

Although the viscosity of heavy oil decreases after aquathermolysis processes, secondary polymerization processes of reactive compounds in their products may occur, leading to an increase in the content of resinous–asphaltene substances and an in viscosity [24]. Hydrogen donors, such as tetralin, can close the active chains formed during the aquathermolysis of heavy oil (shown as Scheme 1):

$$\text{Heavy Oil + water} \xrightarrow{\text{aquathermolysis}} \text{light Hydrocarbons +}$$

emitted gases (CH₄, CO, H₂) + active chains

$$\text{CO} + H_2O \longrightarrow CO_2 \ + \text{active hydrogen «H}_2\text{»}$$

| active hydrogen «$H_2$» | + | active chains | $\longrightarrow$ | active chains reactions (low molecular mass) |

| active chains | + | active chains | $\longrightarrow$ | active chains reactions (low molecular mass) |

**Scheme 1.** The role of water in hydrogenation of active chains.

The hydrogen donor additive (tetralin) present in the reaction system is capable of releasing hydrogen free radicals, which can react with active chains formed during aquathermolysis and inhibit polymerization. The reaction can be expressed as follows (Scheme 2):

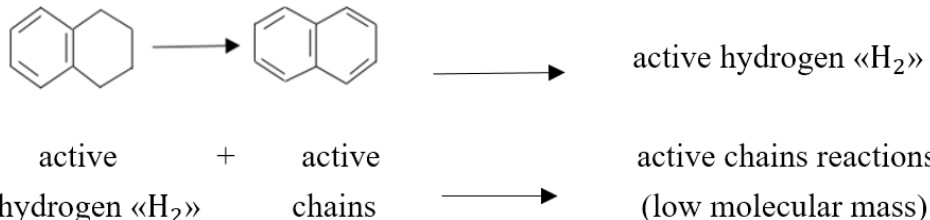

| active hydrogen «$H_2$» | + | active chains | $\longrightarrow$ | active chains reactions (low molecular mass) |

**Scheme 2.** The role of tetralin in hydrogenation of active chains.

At the same time, since the size of the active hydrogen molecule is small, the rate of its transfer is high and leads the active chains in heavy oil quickly ending; therefore, in the presence of a hydrogen donor, the viscosity of heavy oil after the reaction does not increase [24].

The method of fractionation of complex mixtures of high molecular mass compounds has been widely used for a long time [27].This method is intended for the separation of

multicomponent mixtures into narrow fractions with a homogeneous composition to reveal the physicochemical characteristics and their differences [28].

Liquid adsorption column chromatography is a widely used method for fractionation of resins [29–31].

In the liquid adsorption chromatography method used in [29], eight fractions of resin with different solvents were obtained: n-heptane/toluene 80:20 (fraction 1); n-heptane/dichloromethane 90:10 (fraction 2), 85:15 (fraction 3), and 80:20 (fraction 4); and dichloromethane/methanol 95:5 (fraction 5), 90:10 (fraction 6), 80:20 (fraction 7), and 75:25 (fraction 8).

Furthermore, reversed-phase chromatography and high-performance liquid chromatography methods have been used in addition to liquid adsorption chromatography [32,33]. In [34], the authors obtained four resin fractions using a mixture of the solvents heptane/benzene, benzene, and benzene/ethanol and liquid adsorption chromatography on ASK silica gel. The authors of [30] obtained four fractions of resins (40:60, 60:40, 80:20, and 100) by adding different amounts of acetone to toluene. Separation of resins into narrow fractions with different molecular masses and polarities makes it possible to separate a fraction that has properties closer to the hydrocarbon fraction (saturated with aromatics) and, by using fractionation of complex mixtures of high molecular weight heteroorganic compounds, it is possible to study the compositions and structures of the molecules of "native" asphaltenes and resins of oils of different natures in more detail.

The aim of this study was to establish the nature of structural group changes in resin molecules in the process of catalytic in situ upgrading of high-viscosity oils.

## 2. Results and Discussion

Some of the possible mechanisms for the decomposition of the precursor cobalt-containing catalyst are as follows:

1.  The precursor decomposes to cobalt carbonate with further decomposition to cobalt oxide under the influence of temperature. After that, the cobalt oxide (II) interacts with hydrogen sulfide to form cobalt sulfide particles. Furthermore, these particles are involved in catalytic conversion and upgrading of the composition of high-viscosity oil (Scheme 3) [35].

$$(RCOO)_2Co \xrightarrow{t} CoCO_3 + RC(O)R$$

$$CoCO_3 \longrightarrow CoO + CO_2$$

$$CoO + H_2S \longrightarrow CoS + H_2O$$

$$R\text{-}S\text{-}R \xrightarrow[t]{CoS} RH + R'SH$$

**Scheme 3.** Possible mechanism for the decomposition of the precursor through carbonate stage.

2.  Cobalt hydroxide forms at the same time as further decomposition to cobalt oxide and subsequent interaction with hydrogen sulfide (Scheme 4).

As shown in Figure 1, most of the resin fractions are concentrated in the R1 and R3 fractions (75% isooctane +25% benzene and 75% isooctane +25% isopropyl alcohol, respectively). Furthermore, during the experiments, the redistribution of components between the resin fractions and the decrease of resins content was observed in the process of catalytic aquathermolysis, mainly for the R1 and R2 fractions.

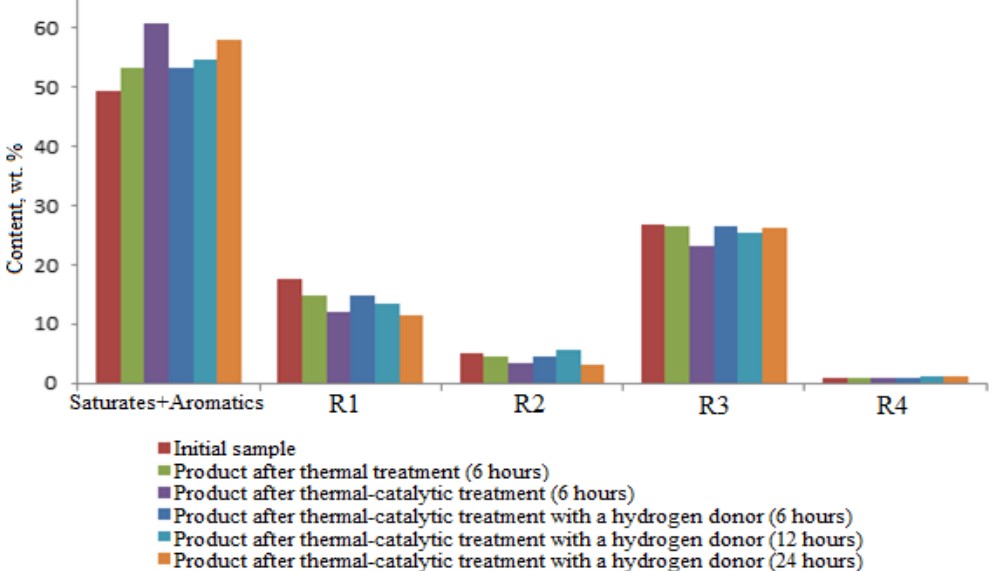

**Scheme 4.** Possible mechanism for the decomposition of the precursor through hydroxide stage.

**Figure 1.** Fractional composition of oil resins from Ashalcha field.

Figures 2–4 present the results of the MALDI spectroscopy of the resin fractions of the initial oil and after catalytic exposure with the hydrogen donor for 6 h. The molecular mass noticeably increases with the separation of fractions from R1 to R4.

The results of the MALDI spectroscopy showed that the molecular masses of all resin fractions decreased after catalytic exposure, mainly with a hydrogen donor. In the R1 fractions, the transition from control experiment to catalytic exposure with a hydrogen donor showed a decrease in the intensity of the fragmentation ions and a shift of the ion peaks to the left, into the lower molecular masses, which was the result of the redistribution of high molecular precipitation ions to light fractions (saturated and aromatic hydrocarbons).

The results of the elemental composition indicated a similar effect for the catalyst as in the case of asphaltenes [13]. The mass fraction of light fractions (saturated and aromatic hydrocarbons) increased as a result of removing alkyl substituents in resins and asphaltenes, respectively; the H/C ratio for resins decreased (Table 1).

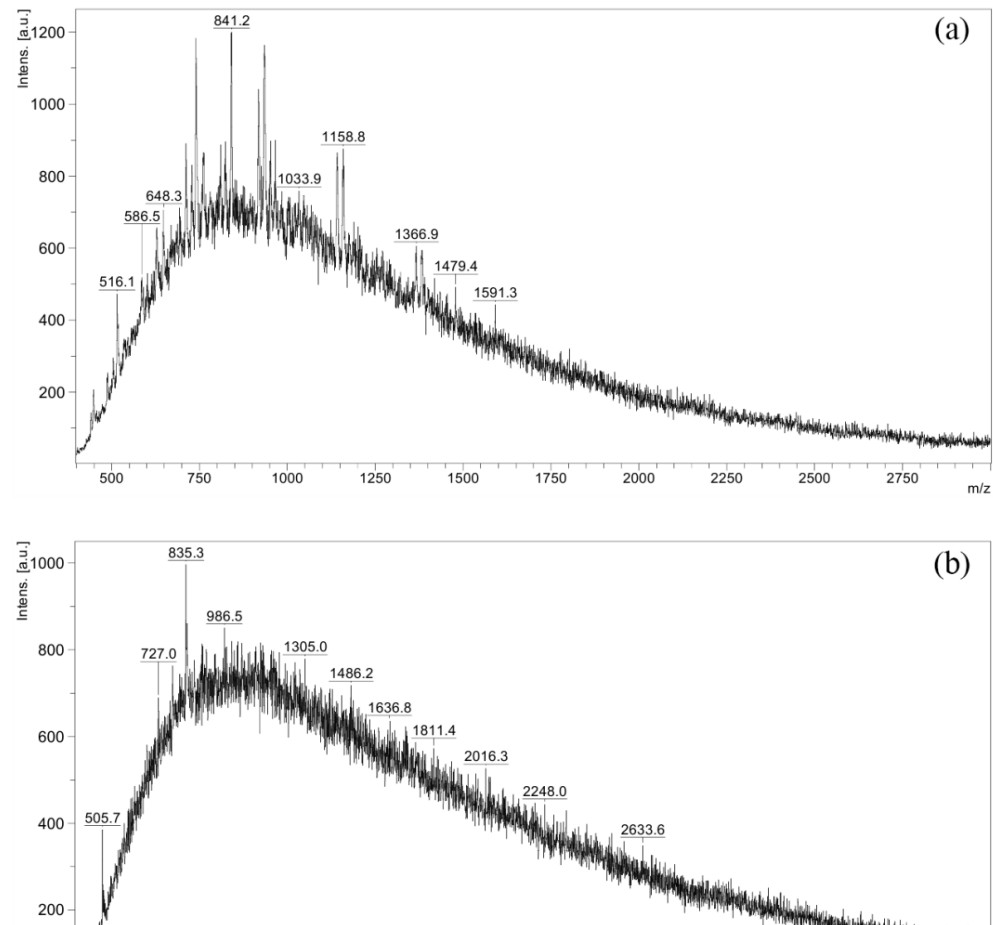

**Figure 2.** MALDI mass spectra of the initial oil resins: (**a**) R2; (**b**) R3.

**Table 1.** Elemental composition of resins fractions.

| Resins Fraction | Content, Mas.% | | | | | |
|---|---|---|---|---|---|---|
| | C | H | N | S | O | H/C |
| Initial oil | | | | | | |
| R1 | 83.64 | 10.88 | 0.02 | 2.18 | 3.28 | 1.562 |
| R2 | 83.04 | 10.14 | 0.23 | 2.64 | 3.95 | 1.467 |
| R3 | 75.29 | 9.19 | 0.95 | 5.83 | 8.74 | 1.466 |
| R4 | 82.28 | 9.51 | 1.02 | 2.88 | 4.31 | 1.388 |
| Non-catalytic aquathermolysis product (6 h) | | | | | | |
| R1 | 83.8 | 10.68 | 0.17 | 2.14 | 3.21 | 1.531 |
| R2 | 83.36 | 10.19 | 0.26 | 2.48 | 3.71 | 1.468 |
| R3 | 71.98 | 8.85 | 0.84 | 7.33 | 11.00 | 1.477 |
| R4 | 81.18 | 9.69 | 0.95 | 3.27 | 4.91 | 1.434 |
| Catalytic aquathermolysis product with a hydrogen donor (6 h) | | | | | | |
| R1 | 82.38 | 10.76 | 0.12 | 2.83 | 3.91 | 1.569 |
| R2 | 76.47 | 9.05 | 0.18 | 6.01 | 8.29 | 1.421 |
| R3 | 74.54 | 8.86 | 1.02 | 6.54 | 9.04 | 1.428 |
| R4 | 80.11 | 9.26 | 1.00 | 4.04 | 5.59 | 1.388 |

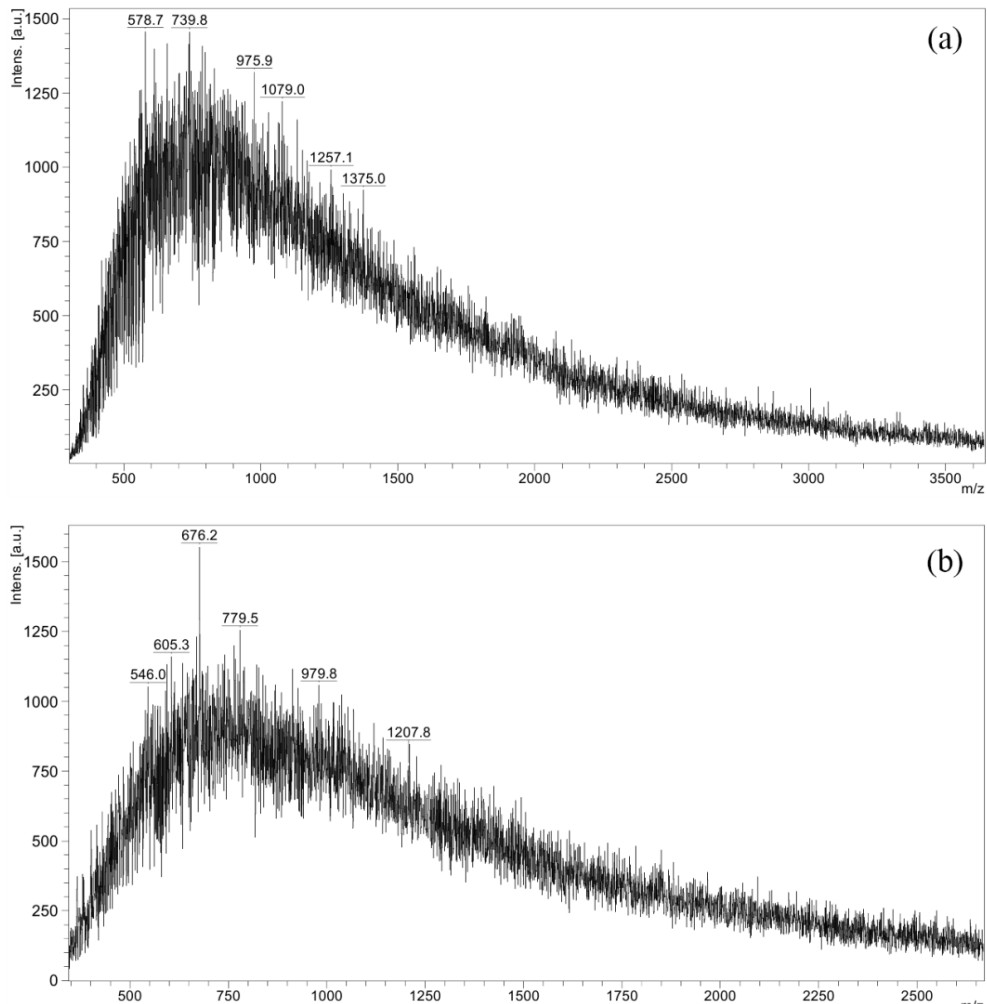

**Figure 3.** MALDI mass spectra of the control experiment resins (6 h): (**a**) R2; (**b**) R3.

The results of $^1$H NMR fully confirmed the catalytic effect of the redistribution of hydrogen atoms in resin molecules. Furthermore, the proportion of aromatic protons that were part of the low-molecular resin molecules R1 and R2 increased under conditions of intensification of destructive hydrogenation, mainly along the least strong carbon–heteroatom bonds in oils after thermal steam treatment (TST) in the presence of a catalyst and a hydrogen donor (Figure 5).

In contrast, at high-molecular parts of the resins R3 and R4, an increase in the aliphatic hydrogen index was observed as a result of the redistribution of part of the low-molecular resin molecule fractions, which were separated from the side substituents of the aromatic rings. Thus, the data obtained indicate the antibatic nature of the change in the aromatic factor of resin molecules in catalytic processes.

Table 2 shows the SGA data for fractions of Ashalcha oil and resins, as well as for the products of non-catalytic and catalytic aquathermolysis. Figure 6 shows the spatial structures of the resin fraction molecules of the initial oil.

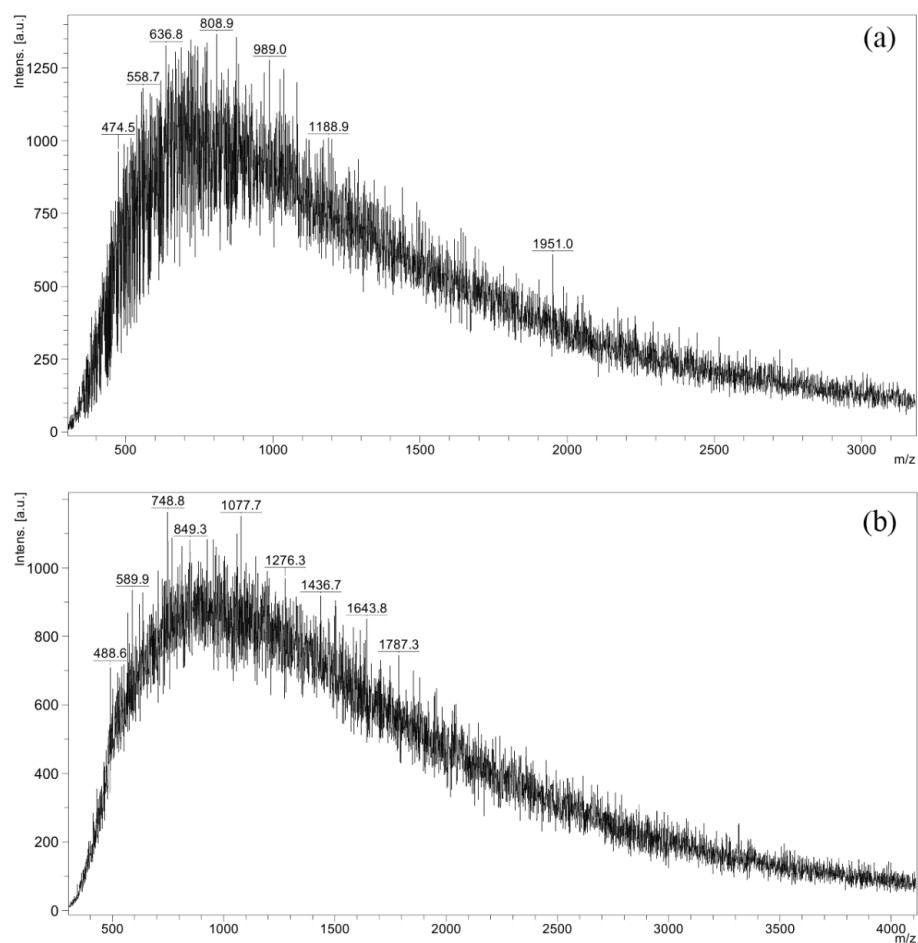

**Figure 4.** MALDI mass spectra of oil resins after catalytic exposure with the hydrogen donor (6 h): (**a**) R2; (**b**) R3.

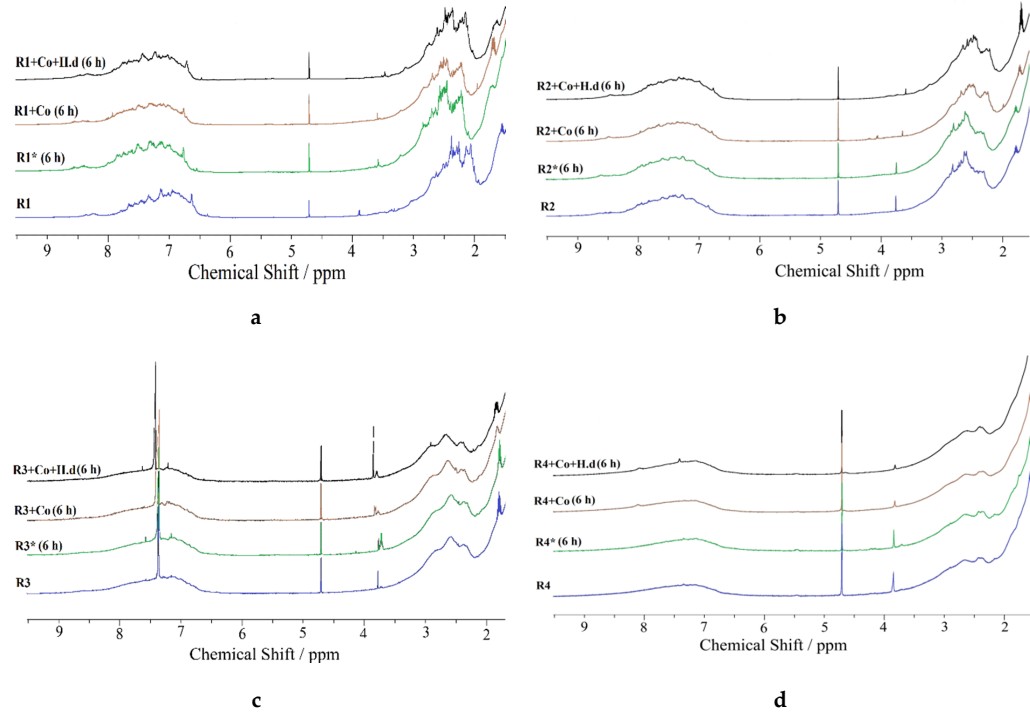

**Figure 5.** $^{1}$H NMR spectra of resin fractions: (**a**) R1; (**b**) R2; (**c**) R3; (**d**) R4.

**Table 2.** Average physicochemical and structural parameters of the resin fraction molecules from the Ashalcha initial oil field and for the products of non-catalytic aquathermolysis.

| Parameters | Initial Oil | | | | Catalytic Aquathermolysis Product with a Hydrogen Donor (6 h) | | | |
|---|---|---|---|---|---|---|---|---|
| | R1 | R2 | R3 | R4 | R1 | R2 | R3 | R4 |
| MM | 900 | 941 | 987 | 1080 | 698 | 710 | 849 | 1245 |
| Number of atoms in an average molecule | | | | | | | | |
| C | 62.7 | 65.1 | 61.9 | 74.0 | 47.9 | 45.2 | 52.7 | 83.1 |
| H | 97.1 | 94.7 | 90.0 | 101.9 | 74.5 | 63.7 | 74.6 | 114.4 |
| N | 0.01 | 0.2 | 0.7 | 0.8 | 0.06 | 0.09 | 0.6 | 0.9 |
| S | 0.6 | 0.8 | 1.8 | 1.0 | 0.6 | 1.3 | 1.7 | 1.6 |
| O | 1.9 | 2.3 | 5.4 | 2.9 | 1.7 | 3.7 | 5.1 | 4.3 |
| Number of rings | | | | | | | | |
| $K_t$ | 5.5 | 6.2 | 6.2 | 11.8 | 3.5 | 5.6 | 5.9 | 11.5 |
| $K_a$ | 4.0 | 5.4 | 5.3 | 5.6 | 3.2 | 3.7 | 4.7 | 7.3 |
| $K_{sat}$ | 1.5 | 0.8 | 0.9 | 6.2 | 0.3 | 1.9 | 1.2 | 4.2 |
| $m_a$ | 1.72 | 2.05 | 2.01 | 2.10 | 1.83 | 1.64 | 1.87 | 2.48 |
| $\sigma_a$ | 0.50 | 0.55 | 0.58 | 0.59 | 0.45 | 0.54 | 0.53 | 0.59 |
| Distribution of carbon atoms, % | | | | | | | | |
| $f_a$ | 28.9 | 36.6 | 32.8 | 31.1 | 31.9 | 33.6 | 34.1 | 35.6 |
| $f_n$ | 8.9 | 4.1 | 5.2 | 34.2 | 1.6 | 15.8 | 8.3 | 20.0 |
| $f_n$ | 62.2 | 59.2 | 62.0 | 34.7 | 66.5 | 50.6 | 57.6 | 44.4 |
| Number of different types of carbon atoms in an average molecule | | | | | | | | |
| $C_a$ | 18.1 | 23.8 | 20.3 | 23.1 | 15.3 | 15.2 | 18.0 | 29.6 |
| $C_n$ | 5.6 | 2.7 | 3.2 | 25.3 | 0.8 | 7.1 | 4.3 | 16.6 |
| $C_p$ | 39.0 | 38.6 | 38.4 | 26.7 | 31.8 | 22.9 | 30.4 | 36.9 |
| $C_\alpha$ | 6.8 | 9.5 | 8.1 | 9.7 | 5.6 | 6.0 | 6.62 | 11.95 |
| $C_\gamma$ | 8.6 | 3.1 | 3.1 | 3.1 | 7.0 | 4.6 | 1.7 | 3.1 |

Here, MM is the molecular mass; the number of carbon atoms in the middle molecule is indicated by $C_a$ for aromatic rings, $C_n$ for naphthenic rings, $C_p$ for aliphatic fragments, $C_\alpha$ for the $\alpha$-position of the aromatic ring, and $C_\gamma$ for the $\beta$-position and further from the aromatic ring; the number of rings is indicated by $K_t$ (total), $K_a$ (aromatic), and $K_{sat}$ (saturated); the fraction of carbon atoms is indicated by $f_a$ (aromatic fragments), $f_n$ (naphthenic fragments), and $f_p$ (paraffin fragments); $\sigma_a$ is the degree of substitution of aromatic nuclei; and $m_a$ is number of blocks in a molecule.

The resinous substances R1 isolated from the initial oil were a mixture of heteroorganic, mainly monoblock, molecules with an average composition of $C_{62.7}H_{97.1}N_{0.01}S_{0.6}O_{1.9}$ and an average molecular mass of 900 amu, while the proportion of monoblock molecules was 28% ($m_a$ = 1.72). In the structural blocks of resin molecules there were 30–36 carbon atoms, on average, combined into predominantly pentacyclic ($K_t$* = 3.2) naphthenic aromatic structures, including bi- and, in at least 33% of the cases, tricyclic aromatic nuclei ($C_a$* = 2.33). In aromatic structures, molecules were found to concentrate 28.9% of the total number of carbon atoms. For each aromatic nucleus in the blocks of molecules of these resins, there were 0.9 naphthenic rings on average. From the $C_p$* = 22.7 paraffinic atoms in the middle block, five were included in the methyl substituents at the alkyl and alicyclic structures of the molecules ($C_\gamma$* = 5.0), which indicated that long alkyl chains of linear or weakly branched structures were widespread in the molecules. As the available data show (Table 1), 99% of the molecules of the resin fraction R1 did not contain nitrogen atoms, while 40% did not contain sulfur. Also, nitrogen was not present in 99% of the structural blocks in the resinous molecules and 65% of the blocks (N* and S* = 0.01 and 0.35, respectively) were insoluble. However, the concentration of oxygen in the resins was very high. They contained widespread molecules with about two oxygen atoms (O).

Two oxygen atoms were present in no less than 10% of the structural blocks (O* = 1.10). This indicated that the resins contained a significant number of carboxylic acids and ester groups linking the structural blocks of molecules.

**Figure 6.** Spatial structures of resin fractions of initial oil.

In this work, we deployed a resin separation scheme that led to the isolation of fractions from R1 to R4 in the order of a gradual increase in the average polarity (absorbability) of the molecules. In parallel, the average sizes of the latter also increased significantly (the measured values of molecular mass increased from 900 to 1080 amu). In the R2 fraction compounds of the general composition $C_{65.1}H_{94.7}N_{0.2}S_{0.8}O_{2.3}$ accumulated, with an average molecular mass of about 941 amu. The molecules of this fraction were two-block molecules (ma = 2.05). The role of the aromatic nuclei in 37% of the blocks was played by naphthalene rings. Among the rest, the most common were probably anthracene structures ($K_a$* = 2.63). Monocyclane fragments were condensed with these nuclei ($K_n$* = 0.39).

Aliphatic structures contained 59.2% of the total number of carbon atoms in the resin fraction R2. The average number of paraffin C-atoms in the structural block in this fraction approached 19 ($C_p$*) and the average number of such atoms in the molecules as a whole was 39. However, the number of methyl substitutes in the average block here was less than 2 ($C_\gamma$*=1.5). Nitrogen atoms were present in only 20% of the R2 molecules (approximately 10% of the total number of structural blocks), and sulfur atoms were present in 80% of the molecules or 39% of the blocks. The oxygen in the R2 fraction was very high, indicating that this fraction contained almost huge amounts of carboxylic acids and their esters.

All the structural characteristics of the carbon skeletons of the molecules isolated as the polar isooctane–isopropyl fraction R3 ($C_{61.9}H_{90}N_{0.7}S_{1.8}O_{5.4}$) were close to those of the R2 components described above. However, there were significant differences in their hetero-element compositions, such as for sulfur, nitrogen, and oxygen, which were two times larger than for R2 (S = 1.8, S* = 0.9; N = 0.7, N* = 0.35; O = 5.4, O* = 2.7), and, of course, this fraction was highly polar and highly adsorbed.

The chloroform fraction R4 ($C_{74}H_{101.9}N_{0.8}SO_{2.9}$) also accumulated resinous substance, but not in such quantities as in R3. Actually, sulfur atoms occurred on average in almost

half of the structural blocks. There were also significant differences in the carbon skeletons of these resinous fractions. Among the aromatic nuclei underlying the structural blocks of the R4 molecules, there were more that were bicyclic and tricyclic (higher than $K_a^* = 2.67$). Particularly significant differences between the R3 and R4 fractions were found in the very high prevalence of alicyclic structures. Moreover, comparing R3 ($K_n^* = 0.45$) and R4, for the molecules with aromatic nuclei, the latter's blocks had a higher average of three naphthene rings ($K_n^* = 2.95$).

Figure 7 shows the spatial structures of resin molar fractions of the catalytic aquathermolysis product (experiment duration: 6 h).

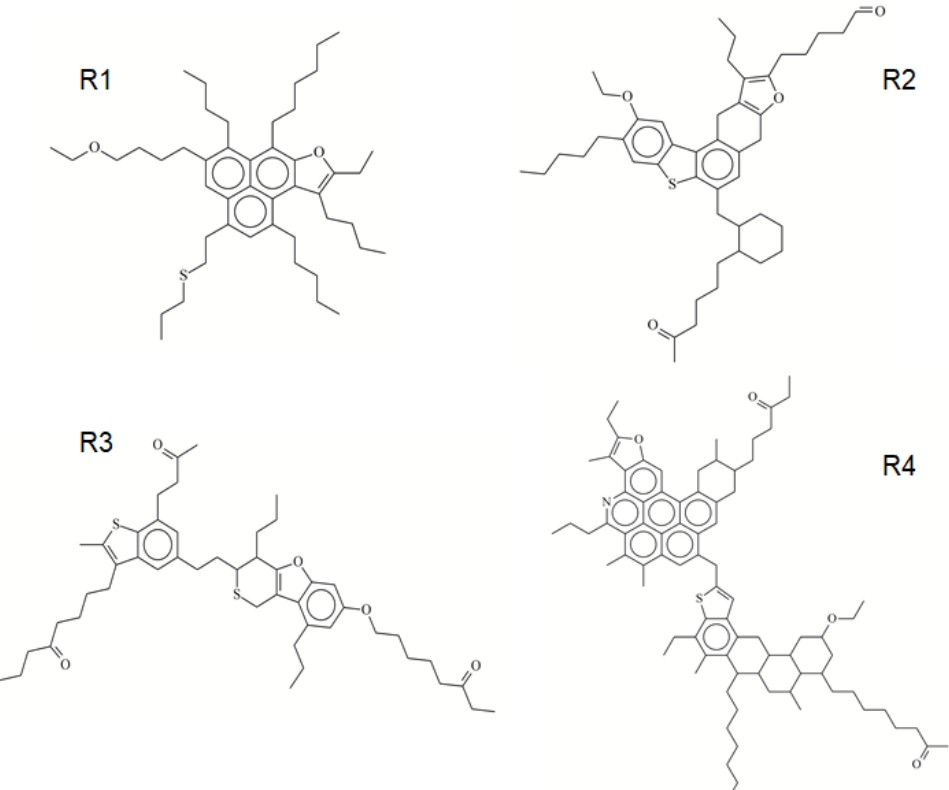

**Figure 7.** Spatial structures of the resin fractions of the catalytic aquathermolysis product with a hydrogen donor (6 h).

Compared to the resins of the initial oil, the separated resin fractions resulting from thermal steam treatment with a catalyst based on cobalt and a hydrogen donor were significantly different. The fraction of the R1 resins of the catalytic aquathermolysis product consisted of heteroorganic monoblock and double-block molecules with an average composition of $C_{47.9}H_{74.5}N_{0.06}S_{0.6}O_{1.7}$ and an average molecular mass of 698 amu. Moreover, the molecular mass noticeably decreased in contrast to the initial oil, where the share of monoblock molecules was 17 % ($m_a = 1.83$). The structural blocks of resin molecules contained 26–33 carbon atoms, on average (3–4 less than in the initial oil), combined into naphthenic aromatic structures, including monocyclic and, in no less than 75% of cases, bicyclic aromatic nuclei ($K_a^* = 1.75$), while in aromatic structures, molecules were found to concentrate 31.9% of the total number of carbon atoms. Each aromatic nucleus in the molecular units of these resins had an average of 0.2 naphthenic rings; i.e., almost five times fewer than the initial oil. From the $C_p^* = 17.4$ paraffinic atoms in the middle block, three were included in the methyl substituents at the alkyl and alicyclic structures of the molecules ($C_\gamma^* = 3.8$), which confirmed the catalytic effect of breaking bonds in the side chains of naphthenic aromatic structures. As the data show (Table 1), 94% of the resin fraction molecules R1 did not contain nitrogen atoms, and the same applied to the initial oil,

40% of which did not contain sulfur. Furthermore, nitrogen was absent in 97% of the structural blocks in resinous molecules, and 67% of the blocks were sulfurless ($N^*$ and $S^* = 0.03$ and 0.33, respectively). The catalyst promoted desulfurization reactions in a 6 h experiment. It is likely that much longer treatment times would lead to even greater desulfurization. The concentration of oxygen in the resins after the catalytic aquathermolysis was lower than in the initial oil resins. Almost all structural blocks had one oxygen atom ($O^* = 0.93$). This indicated that, under the action of a catalyst and a hydrogen donor, not only were the C-S bonds broken, but also the C-O bonds of carboxylic acids and ester groups.

In the R2 fraction there were compounds of the general composition $C_{45.2}H_{63.7}N_{0.09}S_{1.3}O_{3.7}$ with an average molecular mass of about 710 amu, and the molecules of this fraction were a monoblock and double-block mixture ($m_a = 1.64$). Aliphatic structures contained 50.6% of the total number of carbon atoms in the resinous fraction R2. The average number of paraffinic carbon atoms in the structural block in this fraction approached 14 ($C_p^*$), and the average number of such atoms in molecules as a whole was 23, which indicated the detachment of the side alkyl chains of the aromatic rings and redistribution towards lighter fractions. Unlike the initial oil, the number of methyl substituents was greater than 2 ($C_\gamma^* = 2.8$) in the middle block. Nitrogen atoms were lower than in the initial oil, and only 9% of R2 molecules (about 5% of the total structural blocks) and 79% of the blocks contained more sulfur atoms. Oxygen in the R2 fraction was as high as in the initial oil.

All structural characteristics of the carbon skeletons of the R3 fraction molecules after 6 h catalytic aquathermolysis with the hydrogen donor ($C_{52.7}H_{74.6}N_{0.6}S_{1.7}O_{5.1}$) were close to those of the R2 components. Compared to the initial oil, the number of rings, both general and aromatic, decreased and the proportion of atoms in naphthenic fragments increased.

In the chloroformed fraction R4, in the product of the catalytic aquathermolysis with a hydrogen donor ($C_{83.1}H_{114.4}N_{0.9}S_{1.6}O_{4.3}$), sulfur atoms were found in, on average, almost 65% of the structural blocks and 36% indicated the presence of nitrogen atoms. As well as for the initial oil, significant differences were also noted in the carbon skeletons of these resinous fraction molecules. Among the aromatic nuclei underlying the structural blocks of the R4 molecules, there were more bicyclic and tricyclic nuclei (higher than the value of $K_a^* = 2.94$). A particularly significant difference between the R3 and R4 fractions was in the very high prevalence of alicyclic structures in them. Compared to R3 ($K_n^* = 0.64$), the R4 molecules with aromatic nuclei in these blocks condensed, on average, about two naphthenic rings ($K_n^* = 1.69$), while in the initial oil there were 3 naphthenic rings in the R4 fraction.

The changes occurring during catalytic aquathermolysis can be described by the following Scheme 5:

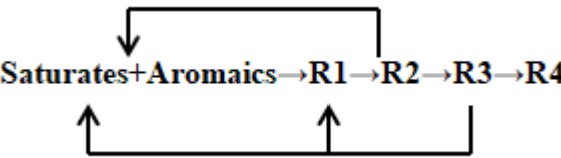

**Scheme 5.** Changes in group chemical composition.

### 3. Research Methods

For this research, we selected the oil of the Ashalcha field from the Permian deposits of the Tatarstan Republic. The physicochemical properties of the initial oil were determined (Table 3).

Ashalcha's oil is heavy (class I—$\rho = 934$–966 kg/m$^3$ (14–20 °API)) and super viscous ($\eta = 1000$–10,000 mPa s), with high sulfur (from 1.81 to 3.5% mass according to GOST 51858-2002).

**Table 3.** - Physical and chemical properties of the initial oil.

| Characteristics | Ashalcha Field |
|---|---|
| Density at 20 °C, kg/m$^3$ | 959.7 |
| Dynamic viscosity, mPa s- at 20 °C | 3306 |
| Elemental composition, mas.% | |
| - carbon | 83.88 |
| - hydrogen | 11.35 |
| - oxygen | 1.2 |
| - sulfur | 3.2 |
| - nitrogen | 0.37 |
| - H/C | 1.62 |
| Component composition, mas.% | |
| - saturated hydrocarbons | 26.33 |
| - aromatic hydrocarbons | 39.55 |
| - resins | 27.37 |
| - asphaltenes | 6.75 |

The catalyst precursor was synthesized according to the following scheme:

At the first stage, the sodium salt of the greasy acid was synthesized by the interaction of distilled tall oil with alkali. The process of saponification of the greasy acid can be described by the following equation (for example, oleic acid):

$$C_{17}H_{33}COOH+NaOH \rightarrow C_{17}H_{33}COONa+H_2O$$

The sodium salt of a greasy acid interacted at heating with a salt of a transition metal (for example, $CoSO_4$):

$$2C_{17}H_{33}COONa+CoSO_4 \rightarrow (C_{17}H_{33}COO)_2Co+Na_2SO_4$$

The donor was Nefras C4-155/205, which is a mixture of naphthenic and aromatic hydrocarbons. It is both a good diluent (it dissolves the polar and non-polar components of oil) and can play the role of a hydrogen donor, which, during cracking, stops the growth of free radicals and prevents their recombination.

Laboratory modeling of the aquathermolysis process was carried out using a high-pressure mixing reactor (300 mL) produced by Párr Instruments (Moline, IL, USA). The model system was a mixture of oil and water with a mass ratio of 70:30. The water–oil emulsion was subjected to temperature exposure (250 °C) in a catalytic process. The cobalt catalyst precursor and the hydrogen donor were introduced at 0.2 mas.% metal and 1.0 mas.% oil, respectively [13]. The working pressure was fixed at 4 MPa. The durations used in the experiment were 6, 12, and 24 h. At the end of the process, the oil was separated from the water by centrifugation on the Eppendorf 5804R (Hamburg, Germany) laboratory centrifuge at 3000 rpm for 1 h.

Fractionation of resins took place as follows: The fractions of maltenes and asphaltenes were obtained from oil samples (5–10 g) after precipitation of asphaltenes with hot isooctane (200 mL) on a paper filter. The fraction of maltenes was separated into oils and four groups of resins using a glass chromatographic column by sequential elution with isooctane (oil), isooctane with benzene in a ratio of 3:1 (R1), isooctane with benzene in a ratio of 1:1 (R2), isooctane with isopropyl alcohol in a ratio of 3:1 (R3), and chloroform (R4) from the adsorbent aluminum oxide (Figure 8).

Matrix-activated laser desorption/ionization (MALDI) was used to determine the molecular masses of resins. When MALDI is used as a method of substance ionization in mass spectrometry, it refers to so-called "soft" ionization methods, which make it possible to ionize large molecules without degradation. The method is based on the use of additional

substance "matrices", the quality of which determines a decrease in the destructive qualities of laser radiation and the ionization of the substance under consideration.

**Figure 8.** Scheme of resins fractionation.

The MALDI mass spectra were obtained with Bruker's ULTRAFLEX III mass spectrometer. The conditions for mass spectra registration were: Nd laser: YAG; wavelength: 355 nm.

Determinations were carried out under vacuum conditions: $10^{-6}$–$10^{-8}$ mbar (in the source: $6.7 \times 10^{-7}$ mbar, in the analyzer: $9.7 \times 10^{-8}$ mbar); temperature in standby mode—room temperature (20 °C), during shooting—500 °C; the mode was linear and a metal target was used without accumulation of mass spectra; 2.5-dihydroxybenzoic acid (DHB) was used as a matrix. In the course of the study, spectra were obtained that represented the Gaussian distribution of the compound molecular mass included in the samples. The average molecular mass of the studied sample was determined at the maximum of the spectra.

The elemental composition (CHNS/O) of oil was determined by burning samples on a CHNS/O analyzer 2400 series II (Perkin Elmer) at a temperature of 975 °C. The CHNS analyzer operating principle is based on the classical technique of burning the sample in an oxygen medium at a temperature of up to 975 °C, while O analysis is a method of recovery in a He/$H_2$ atmosphere. The analysis was carried out on the emitted gases on a katharometer through the method of displacement chromatography, and the separation of gases was carried out on a chromatographic column.

All NMR spectra were obtained using an Avance III Bruker NMR spectrometer (BRUKER BIOSPIN AG, Faellanden, Switzerland), and all measurements were carried out at a temperature of 30 °C. The resin samples were dissolved in deuterated chloroform (CDCl$_3$). The resulting spectra were integrated after baseline correction. For each calculation, the average of the three integration values was taken, while the relative error in the results of the manual integration did not exceed 3 [13].

Structural group analysis (SGA) of the extracted resins was carried out according to the methodology developed at the Institute of Petroleum Chemistry of The Siberian Branch of the Russian Academy of Sciences (SB RAS) [36]. For the SGA, we used the data from the analysis of the elemental composition, average molecular weight, and NMR spectroscopy.

## 4. Conclusions

In summary, the oil resin fractions of the Ashalcha field consisted of a set of molecules with average molecular masses from 900 to 1080 amu, and the resin fractions of the catalytic aquathermolysis product with a 6 h treatment time had molecular masses from 698 to 1245. On average, there was one sulfur atom, one nitrogen atom, and five oxygen atoms per resin molecule. During the transformation, the heteroatom contents (sulfur, nitrogen, and oxygen) decreased in a much larger number of structural blocks of resin molecules under the action of the catalytic agent and the hydrogen donor. Resin molecules were mainly represented by monoblock and double-block structures, with a predominance of aromatic rings over naphthenic ones, with long paraffin chains. As a result of thermal steam treatment with a catalyst and a hydrogen donor, the fraction of atoms in paraffinic and naphthenic fragments decreased, while in aromatic fragments it increased, which confirms the fact that an increase in aromaticity and a decrease in oil aliphaticity take place during the hydrothermal transformations of oil.

**Author Contributions:** Conceptualization, A.V.V.; methodology, I.S.S.S.; investigation, I.Z.R., N.N.S., and G.S.P.; data curation, I.I.M.; writing—original draft preparation, R.K.S. and A.V.V.; writing— review and editing, I.I.M. All authors have read and agreed to the published version of the manuscript.

**Funding:** This research received no external funding.

**Acknowledgments:** This work was supported by the Ministry of Science and Higher Education of the Russian Federation under agreement No. 075-15-2020-931 within the framework of the development program for a world-class Research Center "Efficient development of the global liquid hydrocarbon reserves". The work of Irek I. Mukhamatdinov and Indad Sh.S. Salih was supported by the President of the Russian Federation Grant for Young Russian Scientists MK-1517.2020.3.

**Conflicts of Interest:** The authors declare no conflict of interest.

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
