# Peer review of "Transformation of Resinous Components of the Ashalcha Field Oil during Catalytic Aquathermolysis in the Presence of a Cobalt-Containing Catalyst Precursor"

_catalysts, doi:10.3390/catal11060745_

Round 1

Reviewer 1 Report

In this manuscript, Mukhamatdinov et al. reported the use of a Cobalt-containing catalyst precursor to study the transformation of resinous components of the oil.

In detail, the others used the Ashalcha Field oil to study the aquathermolysis in the presence of a clay catalyst and a hydrogen donor. The authors investigated the effect of duration and steam temperature in catalytic conversion. Further, the products formed under aquathermolysis are characterized using liquid chromatography, MALDI spectrometry, and NMR spectroscopy.

While the work is interesting, several concerns need to be addressed.

  1. It is not very clear in the introduction whether Figures 1-3 are adopted from the literature or own work. Please clarify.
  2. The manuscript lacks complete experimental details and data. 
  3. There are no details on how the liquid chromatography data was collected, and sample preparation was carried out and analyzed.
  4. The authors have not provided any 1H NMR data and elemental analysis data.
  5. It is essential to provide all 1H NMR data and support it with spectra, also MALDI and LC spectra. I suggest the authors offer complete and detailed supporting information.

Author Response

1. Figures 1-3 are taken from literature source : [21] Katritzky, A.R.; Nichols, D.A.; Siskin, M.; Murugan, R.; Balasubramanian, M. Reactions in high-temperature aqueous media. Chem. Rev. 2001, 101, 837–892.

2. The manuscript describes in detail the experimental part and how resin fractions MALDI spectroscopy, elemental analysis, NMR spectroscopy and structure group analysis were carried out.

3. The chromatography technique is described in section 3 of the paper on page 13.

4. We have provided 1H NMR data and elemental analysis data.

5. We have provided data of MALDI.

Reviewer 2 Report

Although this manuscript frames a topic of energy interest, is well designed and reaches interesting conclusions regarding the transformation of heavy oil and the different composition and properties of the fractions obtained, it is not focused from a catalytic point of view. That is, there is a lack of focus from the introduction to the experimental details, eg. it is not described what Co precursors are used, what H donors, how these components are transformed and interact in reaction to produce the different fractions of oil, etc. it is also necessary to carry out the same experiment in the absence of a catalyst and describe how the catalyst influences the breaking of the different bonds, not simply by describing the composition of the end products but the mechanism of the catalyzed reaction, the H-transport, the influence on secondary reactions, etc. In the present form, I think it would be more interesting in other journal like Fuel or Energy and Fuels, etc.

Author Response

We have provided data on the synthesis of cobalt tallate and hydrogen donor, representing naphthene-aromatic compounds.

We have also provided data on MALDI spectroscopy, elemental analysis and NMR spectroscopy of the control experiment.

We have also provided the general scheme of the ongoing transformations. It has been as well explained in details that the main reaction is the transformation of resins through weak carbon-heteroatom bonds.

Reviewer 3 Report

  1. Given that the figures need to be visually clear, the authors should redraw these figures (4 and 7) as they are of poor quality.
  2. Statistics of the results is not described and given in figure 4 and in table 1.       
  3. I would prefer deeper scientific discussion.
  4. Formulas on page 5 are not numbered. 
  5. Where are Tables 2 and 3?
  6. Scheme not mentioned on page 10.

Author Response

  1. Fixed
  2. Fixed
  3. Discussion on the results of MALDI spectroscopy, elemental analysis and NMR spectroscopy is provided.
  4. Fixed
  5. Numbering error. Fixed.
  6. Fixed

Round 2

Reviewer 1 Report

In the revised version of the manuscript, the authors have addressed the comments raised by the reviewer and improved the contents of the manuscript accordingly.  Additionally, the authors have now provided the experimental data on MALDI spectrometry, elemental analysis and NMR spectroscopy along with a proper analysis.  Overall, the revised version of the manuscript have improved significantly and acceptable to be published in Catalyst.

Author Response

We have provided data on the possible transformation mechanism of the precursor of a cobalt-containing catalyst.

Reviewer 2 Report

It is evident that the distribution of aquathermolysis products varies in the presence of Co (catalyst) and hydrogen donors leading to a decrease of the molecular weight by the breakage of the molecular structure trough weaker bonds formed by heteroatoms, but this is the typical behaviour produced by thermal treatments, heteroatoms being evolved mainly in the gas phase together light alkanes and therefore increasing the aromaticity of the rest. As commented in my previous report, there is a lack of information about the state, role and transformations of Co.

Please, compare with previous manuscripts of authors:

2021      Processes 9(1),127, pp. 1-22

2021      Catalysts 11(2),189, pp. 1-22

Even if authors reach similar conclusions (reduction of viscosity and increase of aromaticity) in these manuscripts they focus more attention in the catalytic processes.

Additionally, please, check the similitude of this manuscript, including Scheme 4, between others, with those results previously published by authors in

Journal of Petroleum Science and Engineering 186 March 2020 Article number 106721

Influence of Co-based catalyst on subfractional composition of heavy oil asphaltenes during aquathermolysis

Please, summarize the process indicating the advantages and progress reached in this case.

Author Response

(The authors gave the same response as above.)

Round 3

Reviewer 2 Report

I could say again that the authors have not made any physicochemical characterization of the state, distribution, or changes arising from the proposed catalysts
and that the results and conclusions reached are consistent with those previously published. A previously published figure is eliminated and a paragraph based exclusively on references is introduced. Nothing new.
There is also no clarification of the progress provided by this article regarding the same Co catalysts previously used by the authors. However, as I said from my first review,
the article contains interesting data, although not raised from a catalytic point of view, so I beg you to decide its acceptance based on the opinion of the rest of the reviewers who have already accepted it in its present form. .